# *Pulsatilla koreana* Nakai Extract Attenuates Atopic Dermatitis-like Symptoms by Regulating Skin Barrier Factors and Inhibiting the JAK/STAT Pathway

**DOI:** 10.3390/ijms26072994

**Published:** 2025-03-25

**Authors:** Hye Jin Kim, Musun Park, Seol Jang, Hyun-Kyung Song, Sang Kook Lee, Taesoo Kim

**Affiliations:** 1KM Convergence Research Division, Korea Institute of Oriental Medicine, Daejeon 34054, Republic of Korea; kimhyejin43@kiom.re.kr (H.J.K.); swellseol@kiom.re.kr (S.J.); 2KM Data Division, Korea Institute of Oriental Medicine, Daejeon 34054, Republic of Korea; bmusun@kiom.re.kr; 3Practical Research Division, Honam National Institute of Biological Resources, Gohadoan-gil 99, Mokpo 58762, Republic of Korea; gusrud1654@hnibr.re.kr; 4College of Pharmacy, Natural Products Research Institute, Seoul National University, Seoul 08826, Republic of Korea

**Keywords:** *Pulsatilla koreana* Nakai, atopic dermatitis, JAK/STAT, T helper2 cells, skin barrier

## Abstract

Atopic dermatitis is caused by various factors, including complex interactions between immune responses and imbalances in T helper cells. In order to resolve the side effects of steroid-based treatment and rapidly improve atopy symptoms, the development of preventive substances for new treatments and as food supplements is essential. *Pulsatilla koreana* Nakai (PKN) is traditionally used as an effective herbal medicine for pain relief, anti-inflammation, and edema, and dried PKN is boiled and drunk as a tea to prevent them; however, its effect on skin manifestations such as atopy are unclear. Therefore, we investigated the in vivo and in vitro effects of PKN extract on improving symptoms of atopy as a potential treatment. By evaluating dermatitis scores and conducting histopathological analysis in mice with Dermatophagoides farina-induced atopy-like pathology, we demonstrated that PKN extract alleviated atopy symptoms. Moreover, PKN extract restored a reduction in the protein levels of skin barrier-related factors in skin tissue. Through in vitro analysis, we examined the impact of PKN on JAK/STAT signaling in IL-4/IL-13-stimulated human keratinocytes and elucidated the mechanisms that suppress the levels of skin barrier factors and inflammation. PKN extract inhibited JAK/STAT phosphorylation stimulated by IL-4/IL-13. Furthermore, docking analysis of PKN constituents indicated binding to JNK1/2 and STAT3/6 and a subsequent inhibition of signal transduction. Therefore, this suggests that PKN extract has potential not only as a treatment but also as a food supplement to improve atopic dermatitis by strengthening skin barrier factors and inhibiting key signaling molecules.

## 1. Introduction

Atopic dermatitis (AD) is a common, recurrent, chronic inflammatory and multifactorial skin disease characterized by complex interactions between innate and adaptive immune responses. Numerous studies have focused on the imbalance of various T helper (Th) cells during these immune responses [1,2]. It typically manifests in infancy and childhood; however, the risk of recurrence in adulthood is substantially high. Skin symptoms vary with age; approximately 10–20% of children and 1–3% of adults experience symptoms of AD [3]. Beyond skin involvement, atopy significantly impacts quality of life, potentially leading to mental health issues such as depression and social withdrawal. AD is characterized by symptoms such as skin barrier dysfunction, pruritus, and defects in skin barrier. A compromised skin barrier facilitates the entry of external antigens and allergens, triggering an immune response [4]. This response involves proinflammatory cytokines, which promote the differentiation of naïve CD4+ T cells into Th2 cells and accelerate the secretion of Th2-related cytokines. Notably, IL-4 and IL-13, secreted by Th2 cells, contribute to B cell differentiation and class switching, thus elevating Immunoglobulin E (IgE) levels in serum and promoting eosinophil and mast cell infiltration into the skin. Moreover, these cytokines play pivotal roles in mediating type 2 inflammation during atopic dermatitis, influencing stages of disease progression, including skin barrier dysfunction and immune activation [4,5,6].

The proteins filaggrin, involucrin, and loricrin are essential for keratinocyte differentiation and are crucial to the structure and function of the keratinocyte membrane [7,8]. Therefore, variations in their expression levels indicate skin barrier damage and recovery in atopy. Filaggrin, existing as granules within the stratum corneum, facilitates the formation of connections among fibrous tissues, including keratin in keratinocytes. Involucrin and loricrin, which are integral to the keratinocyte membrane, act as scaffolds for the cross-linking of structural proteins. As keratinocyte differentiation begins, the stratum corneum plays a vital role in skin barrier function, forming the cornified cell envelope [9].

Janus kinases (JAK), a family of intracellular non-receptor tyrosine kinases, transmit cytokine-mediated signals through the JAK-STAT (i.e., Signal transducer and activator of transcription) pathway, playing a crucial role in inflammation signaling. The activation of JAK by transphosphorylation leads to the phosphorylation of STAT, facilitating its dissociation from the receptor. Phosphorylated STATs, either as homodimers or heterodimers, translocate to the nucleus to regulate the transcription of target genes [10,11]. The production of Th2-related inflammatory cytokines is largely dependent on the JAK-STAT pathway. JAK-STAT is involved in the regulation of nerves involved in the regulation of the epidermal barrier and the transmission of itch, as well as various immune pathways associated with AD [10]. Therefore, JAK inhibitors are potential alternatives to the adverse effects associated with systemic immunosuppressants such as steroids and cyclosporine [12,13]. Also, recent clinical trials and studies on AD have reported on topical JAK inhibitors such as ruxolitinib (JAK1/2) and delgocitinib (pan-JAK), as well as oral JAK inhibitors like baricitinib (JAK1/2) and upadacitinib (JAK-1 selective), suggesting the potential of JAK inhibitors as next-generation targeted therapies for AD [12,14].

*Pulsatilla koreana* Nakai (PKN), a perennial plant belonging to the Apiaceae family, is mainly used as an ornamental plant. The name of this plant, whose roots are mainly used as medicines, is Baekduong. PKN thrives not only in Korea but also in northeastern China, where the roots are predominantly used medicinally, and is well known for its effectiveness in relieving pain, inflammation, smallpox, edema, and heart pain. Historical records on herbal medicine from the Chinese Ming Dynasty indicate that PKN may be used as an anthelmintic, with the roots reportedly effective against diarrhea and nosebleeds. To prevent inflammatory diseases such as cardiovascular disease, edema, and vaginitis, the dried roots are sometimes boiled and taken as tea.

Recent studies have also reported on the impact of PKN on allergic diseases such as rhinitis [15,16]; however, research on its potential benefits in AD and skin barrier-related mechanisms remains limited.

Therefore, in this study, we aimed to explore the anti-inflammatory and modulatory effects of PKN extract on skin barrier-related factors in an animal model with Dermatophagoides farinae extract (DfE)-induced AD. Additionally, we investigated the underlying mechanisms using human keratinocyte cell lines to evaluate PKN extract as a potential treatment for AD.

## 2. Results

### 2.1. PKN Extract Improves Atopic Symptoms in DfE-Induced NC/Nga Mice

Initially, we applied DfE to establish an atopy-like symptom model in NC/Nga mice and subsequently evaluated the ameliorative effects of PKN extract (Figure 1A). AD-like lesions were induced by repeated DfE applications on the dorsal region and ears of mice. We observed that symptoms such as edema, maceration, and psoriasis were alleviated with both PKN and dexamethasone (positive control). (Figure 1B). Throughout the experimental period, no significant changes in body weight were observed (Figure 1C). However, the application of DfE resulted in significantly increased ear thickness and dermatitis scores, both of which were reduced in the PKN and positive control groups (Figure 1D,E). Furthermore, spleen weight decreased significantly in the PKN and dexamethasone treatment groups compared with that in the control group, which showed an increase (Figure 1F). These findings demonstrate that PKN extract effectively alleviated symptoms in an animal model with atopy-like symptoms.

### 2.2. PKN Extract Decreases Epidermal Thickness and Mast Cell Infiltration in DfE-Induced AD-like Mice

Increased epidermal thickness and mast cell infiltration are hallmark indicators of atopy onset. We assessed the impact of PKN extract on these markers in DfE-induced NC/Nga mice using H&E and TB staining (Figure 2A). Notably, both epidermal thickness and mast cell infiltration were significantly increased in DfE-induced NC/Nga mice compared with those in normal mice. These decreased in a dose-dependent manner following the administration of PKN (Figure 2B,C).

### 2.3. PKN Extract Suppresses Production of Inflammatory Factors in Serum of DfE-Induced AD-like Mice

An increase in skin thickness, mast cell infiltration, and the serum levels of histamine, IgE, and inflammatory chemo/cytokines are important clinical indicators of atopy. In mice exhibiting DfE-induced AD-like pathology, we observed significant increases in histamine, IgE (Figure 3A,B), and the inflammatory chemokines RANTES, MDC, TARC, MCP-1, MIG, MIL-1α, MIL-1β, and BLC (Figure 3C–J), along with the inflammatory cytokines TNF-α, IL-6, IL-4, and IL-5 (Figure 3K–N), compared to the normal group. Notably, these increases were significantly reduced in both the positive control and PKN treatment groups, with the PKN 300 mg/kg group displaying comparable or superior inhibitory effects on most factors compared with those in the positive control group. These results suggest that PKN extract can alleviate symptoms by suppressing the production of inflammatory chemo/cytokines and exerting anti-histamine and anti-allergy effects.

### 2.4. PKN Extract Restores Decreased Skin Barrier Factors in DfE-Induced AD-like Mice

We evaluated whether PKN extract could restore skin barrier dysfunction in the DfE-treated mouse model by measuring the levels of the skin barrier-related factors filaggrin, involucrin, and loricrin through IHC staining of dorsal skin (Figure 4A). The expression levels of these factors, which decreased following DfE application, were improved in both the positive control and PKN treatment groups (Figure 4B). Additionally, Western blot analysis revealed significant increases in the protein expression levels of these skin barrier factors in the skin tissues of both the positive control and PKN-treated groups (Figure 4C,D). These findings suggest PKN extract may restore the levels of key factors influencing skin barrier integrity.

### 2.5. PKN Suppresses Production of Inflammatory Factors and JAK/STAT Signaling Pathway In Vitro

Following the confirmation of the effect of PKN extract on improving skin barrier factors in tissues, we conducted experiments with human keratinocyte cells to determine whether PKN extract could effectively inhibit related mechanisms. Initially, we administered various concentrations of PKN extract (0–1000 µg/mL) to assess its toxicity toward HaCaT cells (Figure 5A). Next, we investigated the regulatory mechanism of the JAK/STAT pathway mediated by PKN through protein expression analysis. IL-4/IL-13 promoted the phosphorylation of JAK1, JAK2, STAT3, and STAT6 in HaCaT cells, which was significantly suppressed in both the dexamethasone and PKN treatment groups (Figure 5B,C). These results suggest that the modulation of skin barrier and inflammatory factors via the JAK/STAT mechanism is feasible with PKN extract.

### 2.6. Quantitative Analysis of Pulsatilla Koreana Nakai

We quantified seven compounds in PKN using ultra-performance liquid chromatography coupled with triple-quadrupole tandem mass spectrometry (UPLC-TQ-MS/MS). Each compound was identified through comparison with the retention times and m/z values of reference standards, and MS/MS analysis conditions were optimized using multiple reaction monitoring (MRM) modes. We evaluated these compounds in PKN using the established UPLC-TQ-MS/MS method, with the contents detailed in Table 1. Additionally, we provide a chromatogram of these compounds obtained in MRM mode in Figure 6.

### 2.7. Docking Analysis of Seven Compounds

We conducted docking interaction analysis between the seven major compounds found in PKN and JAK1, JAK2, STAT3, and STAT6 (Figure 7). α-hederin exhibited the strongest inhibition of JAK1 and JAK2 (JAK1: −11.522 kcal/mol, JAK2: −10.513 kcal/mol), while hederacoside D and hederasaponin B effectively targeted STAT3 (−10.18 kcal/mol) and STAT6 (−10.745 kcal/mol), respectively, within the JAK/STAT signaling pathway. Additionally, further interaction analyses, beyond the optimal predictions, indicated that other principal compounds of PKN also interact effectively with the JAK/STAT family proteins (Table 2). Visualizations from these analyses revealed that each compound binds to distinct sites within the JAK/STAT proteins, highlighting the specificity of their interactions (Appendix A).

## 3. Discussion

AD, one of the representative allergic diseases alongside asthma, rhinitis, and chronic urticaria, develops due to complex genetic and environmental factors causing immune system malfunctions [17,18]. Atopy primarily manifests as keratinization of the skin, itching, and sleep disturbances, progressing through non-lesional, acute, and chronic stages [19]. Given the chronic nature of atopic dermatitis, which fluctuates between improvement and worsening, long-term management is essential. Therefore, identifying treatments that adjust to the severity and location of lesions and exploring alternatives to long-standing treatments such as topical steroids, anti-histamine agents, intermittent steroid agents, and cyclosporine is crucial due to potential long-term application issues [20,21]. Therefore, we present research supporting the potential of PKN extract as a natural treatment alternative for atopic dermatitis.

Initially, we confirmed the improvement effect of PKN extract in NC/Nga mice. It is difficult for NC/Nga mice to develop atopic dermatitis without specific pathogens, but they are frequently used in atopy research owing to their similarity to human atopic dermatitis lesions under certain triggers such as house dust mites [22]. We established an atopy-like mouse model by applying DfE to the dorsal region and ears of NC/Nga mice, observing improved atopy-like symptoms through clinical indicators such as lichenification, increased ear thickness, and dermatitis score. Further, using H&E and TB staining on the dorsal tissue of the animal model, we noted significant improvements in skin keratin thickness and mast cell infiltration. PKN extract not only enhanced histology but also effectively reduced the serum levels of histamine and IgE, key markers of allergic diseases, which increased upon DfE application.

AD has a complex pathogenesis that is organically linked to the inflammatory response from immune abnormalities within the skin and damage to the external skin barrier. Barrier dysfunction is associated with reduced levels of skin barrier-related molecules such as filaggrin, involucrin, and loricrin [7,23]. Additionally, an imbalance in the immune response owing to disproportionate Th1-derived and Th2-derived cytokines leads to the overexpression of various inflammatory factors in the skin [24,25]. We assessed the effect of PKN extract on restoring skin barrier molecules in DfE-induced atopic-like lesions through immunostaining and protein expression level analysis in dorsal skin tissue. High concentrations of PKN extract significantly increased the expression of filaggrin, involucrin, and loricrin proteins, establishing a foundation for the skin barrier improvement effect of PKN extract.

IL-4 and IL-13 are highly expressed in damaged skin tissues, reducing the expression of skin barrier-related factors. The JAK/STAT signaling pathway plays a crucial role in disrupting immune function in atopy, with IL-4 and IL-13 promoting JAK/STAT phosphorylation [10,26,27]. We validated this mechanism using HaCaT cells, a human keratinocyte cell line, where IL-4/IL-13 stimulation increased the phosphorylation of the JAK1, JAK2, STAT3, and STAT6 genes. Treatment with PKN extract effectively inhibited this phosphorylation, providing robust evidence to support the atopy alleviation effect of PKN extract demonstrated in animal tests.

We identified seven constituent compounds of PKN—Chicoric acid, Anemoside B4, Hederacoside C, Hederacoside D, Hederasaponin B, Anemoside A3, and α-Hederin—using HPLC. Docking analysis predicted that these compounds interact effectively with the JAK/STAT family. A previous study performing large-scale molecular docking analysis identified an alpha level of −8.919 kcal/mol at *p* < 0.05, indicating significant potential for interaction [28]. In our analysis, the interaction probability between PKN-derived compounds and the JAK/STAT series was confirmed to be very high. Furthermore, we discovered that the predicted optimal binding sites for each of the four proteins in the JAK/STAT family vary among the compounds. These results indicate that while a single compound may not interact with all proteins, in the context of the JAK/STAT signaling pathway, various compounds can synergistically mitigate inflammation. This supports the potential of PKN extract as an enhanced treatment for atopic dermatitis.

PKN, a wild perennial herb known for its anti-inflammatory, soothing, and anti-cancer properties, is extensively used in traditional medicine in East Asia, including Korea, to treat bacterial diseases and amoebic dysentery. Notably, it is used in treating ulcerative colitis in traditional medicine, with numerous reports suggesting its potential as an immunomodulatory substance [29,30]. Our research confirmed the atopy alleviation effect of PKN extract through histological and serum analyses and studies on human keratinocyte cell lines using the DfE-induced atopy-like NC/Nga mouse model, establishing the therapeutic potential of PKN extract. Furthermore, our findings are supported by docking analysis of its constituent compounds, which also highlighted the need for further research to elucidate the pharmacological effects and mechanisms of these seven compounds. Although natural products and traditional herbs are crucial alternatives to chemical compounds, their scientific validation remains insufficient. In particular, modern medical research on the efficacy and side effects of PKN is lacking. Most existing studies are based on traditional use cases or animal experiments, and there is a possibility of drug interactions when taken with other medications. Therefore, more safety evaluations and clinical studies are needed. Our findings suggest that PKN extract has potential as a therapeutic agent for skin diseases, demonstrating its immunomodulatory effects and molecular mechanisms.

## 4. Material and Methods

### 4.1. Preparation of PKN Extract

Dried PKN was obtained using low-temperature reflux extraction at 100 ± 2 °C for 3 h using 10 times the DW of the sample. Following extraction, the sample was first filtered using a filtering net and then again through a 53 μm filter. The resulting water extract of PKN was powdered via a water evaporation system equipped with a vacuum pump and rotary evaporator. For in vitro experiments, PKN powder was dissolved in DW and filtered using a 0.2 μm hydrophilic syringe filter. For in vivo experiments, freeze-dried PKN powder dissolved in water was administered orally using a disposable 1 mL syringe.

### 4.2. Animals and DfE-Induced AD Mouse Model

Female NC/Nga mice were acquired from Central Lab Animal Inc. (Seoul, Republic of Korea) and were housed in a pathogen-free (SPF) environment for one week to acclimate them to laboratory conditions (20 ± 2 °C, relative humidity 45 ± 5%, and a 12 h light/dark cycle). The Korea Institute of Oriental Medicine’s Ethics Committee gave its approval for animal experiments (KIOM; approval number: 22-073; approval date: 27 July 2022). For AD induction, DfE ointment was applied, and mice were assigned to one of six random groups: Group 1, normal; Group 2, DfE-induced AD (control); Group 3, positive (DfE + dexamethasone 1 mg/kg p.o) [31]; Group 4, PKN 30 mg/kg (DfE + PKN 30 mg/kg p.o); Group 5, PKN 100 mg/kg (DfE + PKN 100 mg/kg p.o); and Group 6, PKN 300 mg/kg (DfE + PKN 300 mg/kg p.o). To sensitize the skin and temporarily disrupt the skin barrier, 150 μL of 4% sodium dodecyl sulfate (SDS) (Merck, Darmstadt, Land Hessen, Germany) was sprayed on the dorsal skin and ears after hair removal one day prior to application. An hour later, DfE ointment (DfE (Dermatophagoides farinae, Biostir Inc., Osaka, Japan)) was applied twice weekly in doses of 100 mg on dorsal skin and 10 mg on ears to induce atopic-like lesions. After two DfE ointment applications, dexamethasone (Sigma-Aldrich Co., St. Louis, Mo, USA) and PKN extracts (30, 100, and 300 mg/kg) were administered orally over 2 weeks, and phosphate-buffered saline was administered to the normal and control groups. Additionally, neurological and physiological changes that may occur during the treatment process were monitored using indicators such as motor function, food intake, and body weight.

### 4.3. Measurement of Ear Thickness and Evaluation of Dermatitis Severity

Every two weeks, the thickness of the ears was measured with a digital caliper (CAS©, Seoul, Republic of Korea). Using a 0–3 point scale (0 = no symptoms, 1 = mild, 2 = moderate, and 3 = severe) across five AD symptoms (edema, lichenification, erythema/hemorrhage, scarring/dryness, and excoriation/erosion), the degree of dermatitis was evaluated every twice a week for 3 weeks [32,33]. The total cumulative score for these parameters, which are measured twice a week for three weeks, is the dermatitis score.

### 4.4. Histopathological Studies

Post-autopsy, dorsal and ear skin samples were fixed in 10% formalin for at least 24 h, embedded in paraffin, and sectioned into 4 μm slices. Epidermal proliferation was measured using hematoxylin and eosin (H&E; Sigma-Aldrich Co., St. Louis, MO, USA) staining, and the degree of mast cell infiltration was assessed using Toluidine Blue (TB; VB-3013-1, VitroVivo Biotech, Rockville, MD, USA) staining. When quantifying mast cell infiltration, the number of TB-positive cells in four randomly selected field profiles informed the statistical analysis. We performed calculations thrice to determine the density of mast cells per square millimeter using the following formula:No. of mast cells per mm^2^ = Total number of mast cells/Total area (mm^2^).

Images of the skin barrier proteins filaggrin, involucrin, and loricrin were captured using an AX70 microscope (Olympus, Tokyo, Japan) and analyzed with a slide scanner (VMI, Motic, Kowloon, Hong Kong) and automated biometric image analysis software (EBIOGEN, Seoul, Republic of Korea) at 100× magnification.

### 4.5. Immunoassay

Serum was extracted from the hearts of mice that had been given anesthesia, placed in SST tubes (BD Biosciences, San Jose, CA, USA), and centrifuged for 15 min at 4000 rpm after a 2 h pause. Using an enzyme-linked immunosorbent assay (ELISA), we determined the levels of serum IgE (FUJIFILM Wako Shibayagi Corporation, Japan) and histamine (LDN, Nordhorn, Germany). The levels of inflammatory chemokines (RANTES, MDC, TARC, MCP-1, MIG, MIP-1α, MIP-1β, BLC) were quantified using a bead-based immunoassay; LEGENDplex™ Mouse Proinflammatory Chemokine Panel (13-plex) with V-bottom Plate (BioLegend, San Diego, CA, USA). HaCaT cells, pretreated with PKN (50, 100, 300 μg/mL) for 1 h and stimulated with IL-4/IL-13 (10 ng/mL each) (Pepprotech, Thermo Scientific, Waltham, MA, USA) for 24 h. The resultant supernatants and serum were used for further bead-based immunoassays; LEGENDplex™ MU Th2 Panel (TNF-α, IL-6, IL-4, and IL-5) (BioLegend, San Diego, CA, USA). All assays were carried out in accordance with the manufacturer’s guidelines. With the use of a Synergy HTX Multi-Mode Reader (BioTek, Winooski, VT, USA), IgE and histamine absorbance were measured. On a BD LSRFortessaTM Flow Cytometer (BD Biosciences, San Jose, CA, USA), the Mouse Proinflammatory Chemokine and MU Th2 Panels were analyzed using BD CellQuestTM software (version 5.0.1) and LEGENDplexTM Software v8.0 (VigeneTech Inc., Carlisle, MA, USA).

### 4.6. Cell Culture and Cell Viability Assay

In Dulbecco’s modified Eagle medium (DMEM, Gibco) supplemented with 10% fetal bovine serum and 1% penicillin–streptomycin (Thermo Scientific, Waltham, MA, USA), HaCaT cells were grown in 5% CO_2_ at 37 °C. The CellTiter 96 Aqueous One Solution Cell Proliferation Assay (Promega, Madison, WI, USA) was used to measure the vitality of the cells. After 24 h, cells were treated with PKN at doses ranging from 0 to 1000 μg/mL. The cells were seeded at a density of 1 × 10^4^ in 96-well plates. Each well received 20 μL of MTS solution 24 h after the PKN treatment. After three hours, a SpectraMax 340 microplate reader (Molecular Devices, San Jose, CA, USA) was used to measure optical density values at 490 nm.

### 4.7. Sampling and Western Blotting

For the purpose of protein analysis, we extracted tissue sample lysates from 20 mg of dorsal skin in each group. A Bertin Corp. Hard Tissue Homogenizing tube was used to treat the lysates. Tissue protein extraction reagent (T-PER; Thermo Scientific, Waltham, MA, USA) was then used to lyse the samples, and it was supplemented with a protease inhibitor cocktail (Roche, Basel, Switzerland). HaCaT cells were cultured to around 80% confluence, pretreated with PKN for 1 h, and then treated with IL-4/IL-13 (10 ng/mL each) (Pepprotech, Thermo Scientific, Waltham, MA, USA) for 15 min in order to obtain cell lysates for protein analysis. Mammalian protein extraction reagent (M-PER; Thermo Scientific, hWaltham, MA, USA) was used to obtain lysates. Protein concentrations in tissues and cells were measured with Thermo Fisher Scientific’s PierceTM BCA Assay kit. Equal amounts of protein were subjected to 10% SDS-PAGE and transferred to a PVDF membrane using the Trans-Blot Turbo Blotting System (Bio-Rad, Hercules, CA, USA). The membrane was blocked for 30 min using SEA blocking buffer (Thermo Scientific, Waltham, MA, USA), incubated overnight at 4 °C with primary antibodies including those against filaggrin (PA5-115235, Scientific, Waltham, MA, USA), involucrin (ab181980, Abcam, Cambridge, UK), loricrin (55439-1-AP, Pepprotech, Thermo Scientific, Waltham, MA, USA), phospho-JAK1 (#3331S), JAK1 (#3332S), phospho-JAK2 (#3771S), JAK2 (#3230S), phospho-STAT3 (#9131S), STAT3 (#9139S), phospho-STAT6 (#56554S), STAT6 (#9362S), and actin (#4970) (Cell Signaling Technology, Boston, MA, USA). Using Super Signal West Femto Chemiluminescent Substrate (Thermo Scientific, Waltham, MA, USA), proteins were identified following a 1 h incubation period with secondary antibodies coupled with horseradish peroxidase and TBST washing. Using the ChemiDoc Imaging System (Bio-Rad, Hercules, CA, USA) and Image J, respectively, protein levels were seen and measured.

### 4.8. High-Performance Liquid Chromatography (HPLC) Analysis

Ultra-performance liquid chromatography coupled with triple-quadrupole tandem mass spectrometry (UPLC-TQ-MS/MS) analysis was performed using an Agilent 1290 Infinity II system coupled to an Agilent 6495C triple-quadrupole mass spectrometer (Agilent Technologies, Santa Clara, CA, USA). A jet-stream electrospray ionization source was used for mass spectrometry, with separation and detection conditions for analytes established using previously reported methods [34]. Multiple reaction monitoring (MRM) was used to quantify analytes in both positive and negative ion modes. MRM conditions, including retention time, collision energy, and transition, were optimized for all analytes compared to reference standards, and selected parameters are listed in Table 1. Data acquisition was processed using Agilent MassHunter Workstation quantitative analysis software (version 10.1). Chemical standards including chicoric acid, anemoside B4, hederacoside C, hederacoside D, hederasaponin B, anemoside A3, and α-hederin were sourced from ChemFaces (Wuhan, China), and all solvents used were MS-grade, purchased from Thermo Fisher Scientific (Waltham, MA, USA).

### 4.9. Molecular Docking Analysis 

Information on the seven major compounds contained in BDO was confirmed through the PubChem database (https://pubchem.ncbi.nlm.nih.gov/, accessed on 1 April 2024). However, except for chicoric acid, 3D SDF files for six compounds were not supported owing to their large molecular size. Therefore, only the 3D SDF file of chicoric acid derived from the PubChem database was used. Other compounds were converted to 3D SDF files using Isomeric SMILES provided by the PubChem database on the web page, which facilitates the conversion of SMILES to 3D structures (https://www.novoprolabs.com/tools/smiles2pdb, accessed on 1 April 2024). Structures of JAK1, JAK2, STAT3, and STAT6 used in the docking analysis were sourced from the Protein structure provided by the AlphaFold 2.0 database (https://alphafold.ebi.ac.uk/, protein version: v2, accessed on 1 April 2024) [35]. Docking entities were preprocessed by converting the structures to the pdbqt format using the OpenBabel Python library (Python, v3.8.12) [36]. Docking with proteins and compounds was performed using the AutoDock Vina Python library API (v1.1.2) [37] with the following parameters: Exhaustiveness = 100 and Box size = [126]. The remaining parameters were at default values. Exhaustiveness is a parameter that determines how thorough the docking analysis will be calculated. The default value is 8, and researchers can specify a number between 1 and 100 depending on their application. The higher the number, the more rigorous the docking calculation will be. Previous studies have reported that there is little performance improvement when Exhaustiveness is greater than 25, but we set it to 100 for the most accurate analysis. Box_size was chosen as 126, the maximum value provided by the software, to consider all parts of the protein [38]. These results were visualized using AutoDockTools (v1.5.6) [39] and Discovery Studio Visualizer (v21.1.0.20) software. The 3D secondary (ribbon) structure demonstrates the alpha-helix and beta-sheet structures, and the 2D diagram, which enables direct identification of interacting amino acids, was visualized using Discovery Studio Visualizer. AutoDockTools, a method to identify protein pores based on Van der Waals surfaces, was used for 3D molecular surface visualization.

### 4.10. Statistical Analysis

All data are expressed as mean ± standard deviation (SD). To perform multiple group comparisons, a one-way statistical analysis of variance was employed using GraphPad Prism (version 9.5.1, GraphPad Software, Inc., San Diego, CA, USA) software. A value of *p* < 0.05 was defined as statistically significant.

## 5. Conclusions

In conclusion, we demonstrated that PKN water extract alleviated clinical symptoms in DfE-induced NC/Nga atopic-like animals, improved symptoms such as the recovery of skin barrier factors, and effectively inhibited related mechanisms in human keratinocytes. These results support the potential of PKN extract as a treatment for atopic dermatitis and suggest its potential as a candidate to treat various immune- and inflammation-related diseases in the future. In addition, the results of docking interaction analysis of the active compounds constituting PKN suggest that it can potentially be used as an important resource to treat atopy and as an additive in functional foods for the prevention and treatment of atopy.

## Figures and Tables

**Figure 1 ijms-26-02994-f001:**
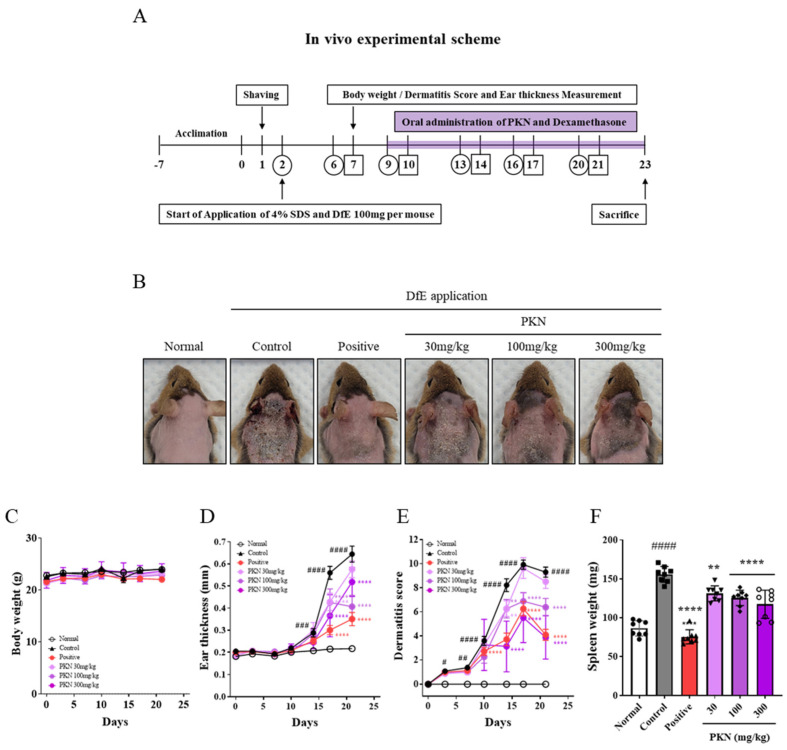
PKN extract improves AD symptoms and reduces dermatitis severity in a DfE-induced AD-like mouse model. (**A**) Experimental design scheme to determine effectiveness of PKN extract in AD-like mouse model. DfE-induced AD-like skin lesions were prevented in PKN groups. Mice were divided into six groups: vehicle, control (DfE treatment only), positive (treated with DfE and oral administration of dexamethasone 1 mg/kg), and PKN at 30, 100, and 300 mg/kg (treated with DfE and oral administration of PKN extract at 30, 100, and 300 mg/kg). (**B**) Clinical symptoms on dorsal skin in each group. (**C**) Body weight, (**D**) ear thickness, (**E**) and dermatitis score were evaluated twice per week for 3 weeks. (**F**) Spleen weight was measured at end of experiment. Values are presented as mean ± SD (*n* = 8). ^####^ *p* < 0.0001, ^###^ *p* < 0.01, ^##^ *p* < 0.005, ^#^ *p* < 0.001 vs. normal group; **** *p* < 0.0001, ** *p* < 0.01 vs. control group. PKN, *Pulsatilla koreana* Nakai; AD, Atopic dermatitis; DfE, Dermatophagoides farinae extract.

**Figure 2 ijms-26-02994-f002:**
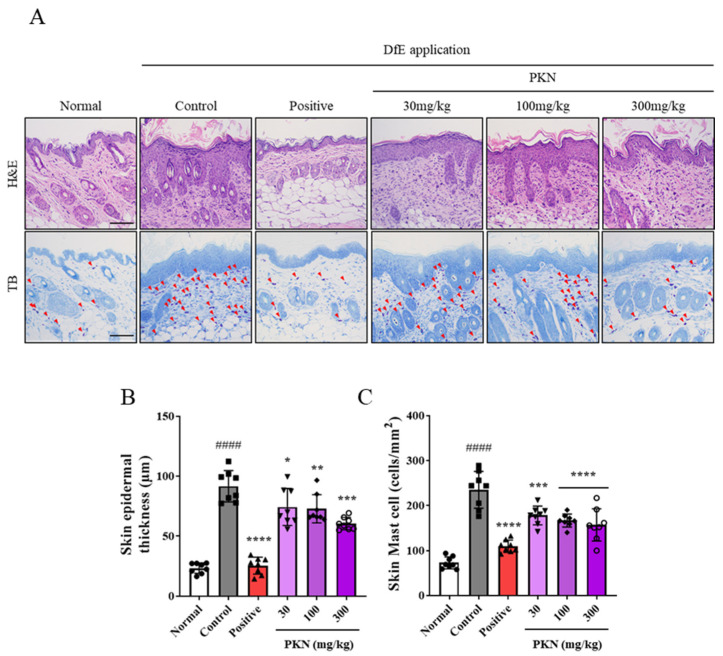
Effect of PKN extract on histopathological features of epidermal thickness and mast cell infiltration assessed by H&E and TB staining. (**A**) Histological analysis using H&E (upper) and TB (bottom) staining of dorsal skin (magnification 100×, scale bar = 100 μm). (**B**,**C**) Statistical analysis of skin epidermal thickness and skin mast cells. Values are presented as mean ± SD (*n* = 8). ^####^ *p* < 0.0001 vs. normal group; **** *p* < 0.0001, *** *p* < 0.001, ** *p* < 0.01, * *p* < 0.05 vs. control group.

**Figure 3 ijms-26-02994-f003:**
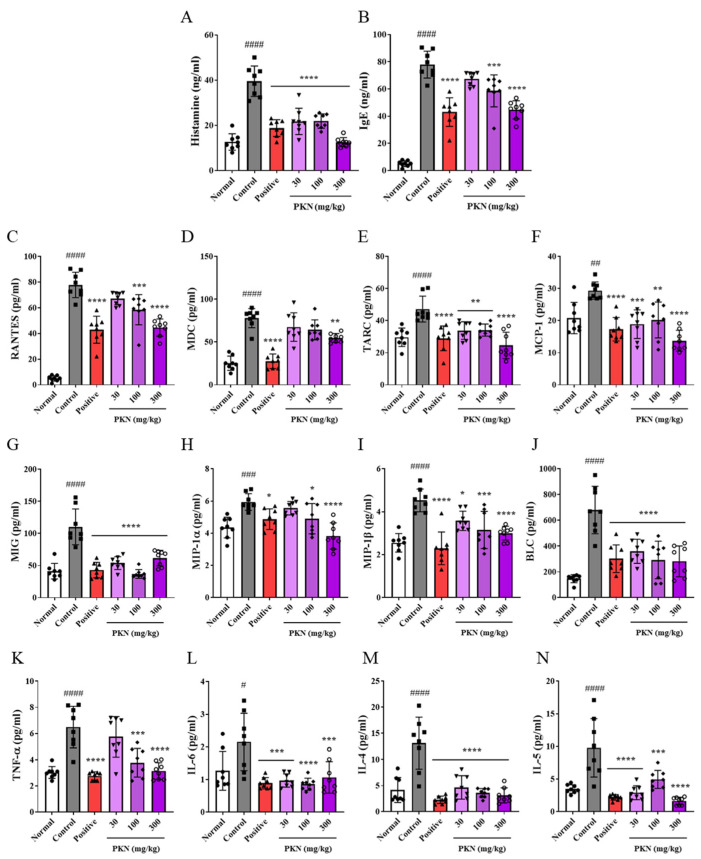
Effect of PKN extract on serum levels of histamine, IgE, and production of inflammatory chemo/cytokines. (**A**) Histamine and (**B**) IgE levels were measured via ELISA. (**C**–**J**) Inflammatory chemokines and (**K**–**N**) inflammatory cytokines were measured via bead-based immunoassay. Values are presented as mean ± SD (*n* = 8). ^####^ *p* < 0.0001, ^###^ *p* < 0.001, ^##^ *p* < 0.01, ^#^ *p* < 0.05 vs. normal group; **** *p* < 0.0001, *** *p* < 0.001, ** *p* < 0.01, * *p* < 0.05 vs. control group.

**Figure 4 ijms-26-02994-f004:**
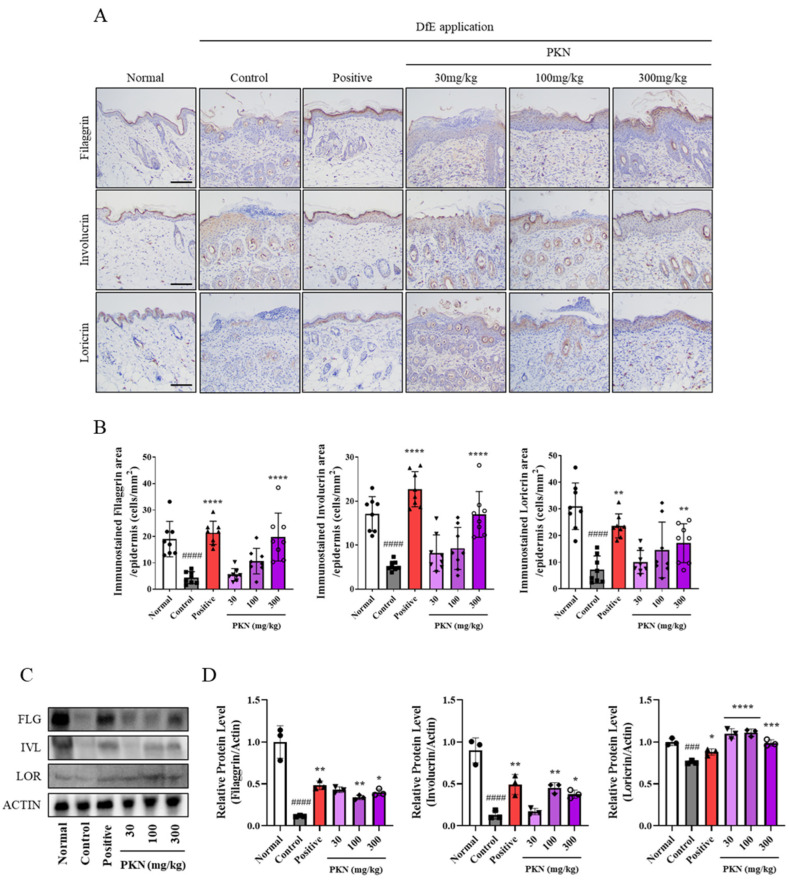
Effects of PKN extract on histopathology and expression of skin barrier proteins in dorsal skin lesions of AD-like mouse model. (**A**) Histological analysis of skin barrier-related factors filaggrin, involucrin, and loricrin using IHC method in dorsal skin of AD-like mouse model (magnification 100×, scale bar = 100 μm). (**B**) Skin barrier proteins. For IHC quantification, epidermal skin area was selected and 3 points per slide were quantified using Image J program (version 1.51). (**C**) Western blot analysis was performed by whole-tissue lysis of mouse dorsal skin. (**D**) Band density was quantified through three independent experiments and normalized total ACTIN. Values are presented as mean ± SD ((**A**,**B**) *n* = 8, (**C**,**D**) *n* = 3). ^####^ *p* < 0.0001, ^###^ *p* < 0.001 vs. normal group; **** *p* < 0.0001, *** *p* < 0.001, ** *p* < 0.01, * *p* < 0.05 vs. control group.

**Figure 5 ijms-26-02994-f005:**
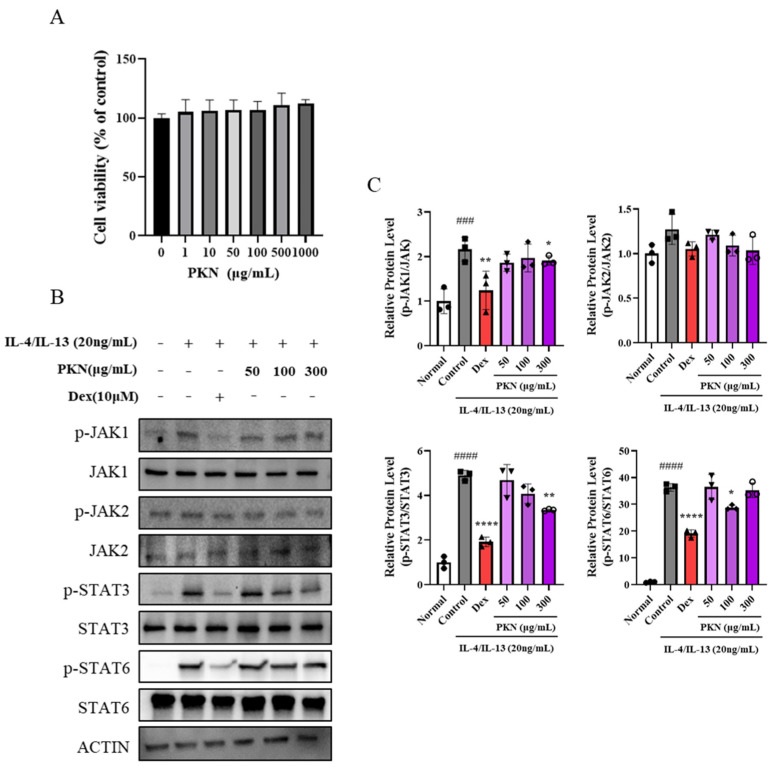
PKN suppresses production of inflammatory factors and JAK/STAT signaling pathway in keratinocyte cells. (**A**) Cell viability of HaCaT cells was evaluated after 24 h exposure to indicated extract concentrations. (**B**) Western blot analysis was performed by whole-cell lysis in IL-4/IL-13-stimulated HaCaT cells. Cells were pretreated with PKN 50, 100, or 300 μg/mL for 1 h and then incubated with IL-4/IL-13 (each 10 ng/mL) for 15 min. (**C**) Band density was quantified through three independent experiments and normalized into total from each gene (JAK1, JAK2, STAT3, and STAT6). Values are presented as mean ± SD (*n* = 3,4). ^####^ *p* < 0.0001, ^###^ *p* < 0.001 vs. control group; **** *p* < 0.0001, ** *p* < 0.01, * *p* < 0.05 vs. IL-4/IL-13-stimulated group.

**Figure 6 ijms-26-02994-f006:**
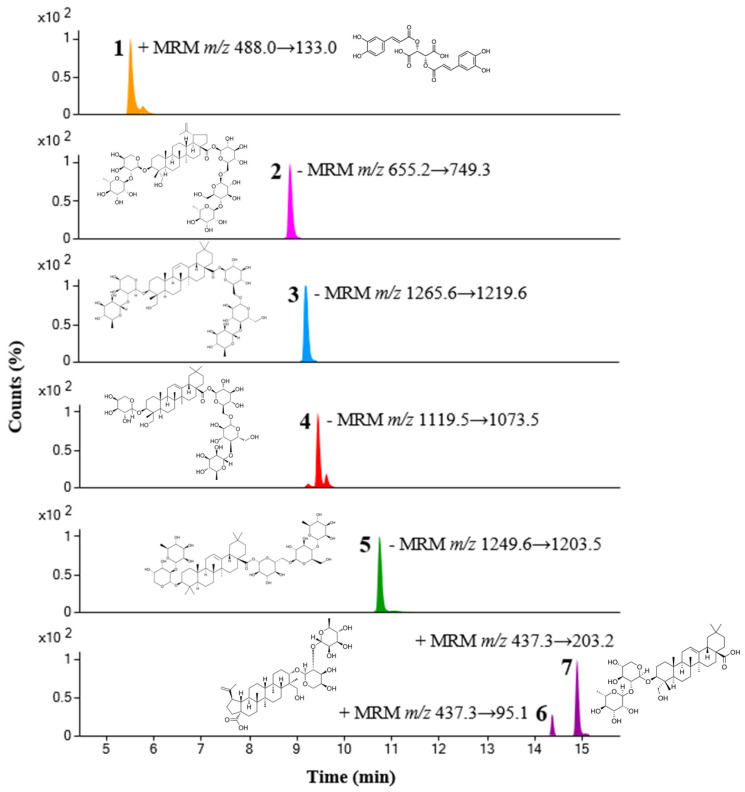
UPLC-TQ-MS/MS analysis MR chromatogram of 7 analytes of PKN: 1, chicoric acid; 2, anemoside B4; 3, hederacoside C; 4, hederacoside D; 5, hederasaponin B; 6, anemoside A3; 7, α-hederin.

**Figure 7 ijms-26-02994-f007:**
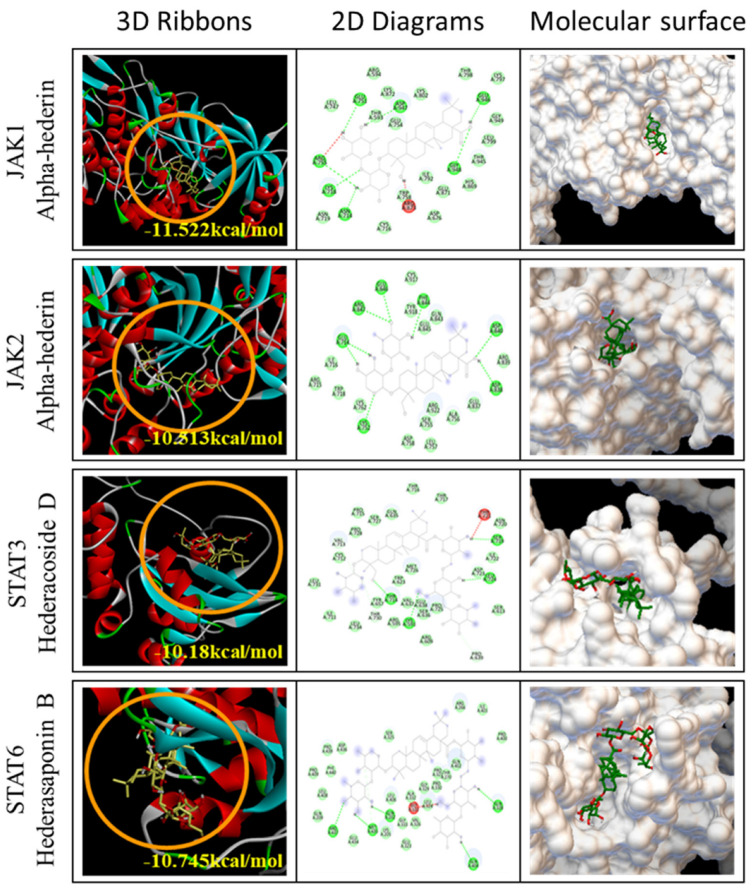
Docking analysis results of the main compounds in the PKN and JAK/STAT family. The major compounds of PKN that best interact at the effective binding sites (orange circles) of JAK1, JAK2, STAT3, and STAT6 protein were visualized, one for each protein. In the 3D ribbon figures, the orange circle is the site where the PKN compound binds, and the yellow text near the circle is the binding affinity value of the compound protein. PKN, *Pulsatilla koreana* Nakai; JAK, Janus kinase; STAT, Signal transducer and activator of transcription.

**Table 1 ijms-26-02994-t001:** Optimized MRM parameters and quantitative results for 7 analytes.

No.	Compound Name	Rt (min)	Ion Mode	Precursor Ion (*m*/*z*)	Product Ion (*m*/*z*)	Collision Energy (V)	Amount (mg/g ± SD)
1	Chicoric acid	5.53	positive	488.0	133.0	38	2.07 ± 0.01
2	Anemoside B4	8.88	negative	655.2	749.3	26	20.68 ± 0.12
3	Hederacoside C	9.22	negative	1265.6	1219.6	22	9.63 ± 0.17
4	Hederacoside D	9.45	negative	1119.5	1073.5	22	1.05 ± 0.02
5	Hederasaponin B	10.75	negative	1249.6	1203.5	22	16.80 ± 0.11
6	Anemoside A3	14.39	positive	437.3	95.1	38	0.38 ± 0.01
7	α-Hederin	14.91	positive	437.3	203.2	22	0.57 ± 0.01

**Table 2 ijms-26-02994-t002:** Docking results between seven compounds of PKN and JAK/STAT family.

JAK/STAT Family	Binding Affinity (kcal/mol)						
	AH	AA3	AB4	CC	HC	HD	HB
JAK1	**−** **11.522**	−9.589	−10.029	−8.741	−10.537	−10.923	−9.915
JAK2	**−** **10.513**	−9.665	−10.122	−9.358	−10.483	−11.147	−10.57
STAT3	−9.232	−8.861	−9.951	−7.819	−9.975	**−** **10.18**	−10.23
STAT6	−10.099	−9.037	−10.373	−8.449	−9.816	−10.648	**−** **10.745**

AH, α-hederin; AA3, Anemoside A3; AB4, Anemoside B4; CC, Chicoric acid; HC, Hederacoside C; HD, Hederacoside D; HB, Hederasaponin B; PKN, *Pulsatilla koreana* Nakai; JAK, Janus kinase; STAT, Signal transducer and activator of transcription. The interaction with the lowest binding affinity for each protein is shown in bold. However, in STAT3, other compounds also interact with the binding site of HD rather than the binding site of HB, so it was judged that HD is likely to have a more effective effect, so HD is indicated in bold.

## Data Availability

Data will be made available on request.

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
