# Peer review of "Pulsatilla koreana Nakai Extract Attenuates Atopic Dermatitis-like Symptoms by Regulating Skin Barrier Factors and Inhibiting the JAK/STAT Pathway"

_ijms, 2025, doi:10.3390/ijms26072994_

Round 1

Reviewer 1 Report

Comments and Suggestions for Authors

Pulsatilla koreana Nakai extract attenuates atopic dermatitis-like symptoms by regulating skin barrier factors and inhibiting the JAK/STAT pathway

The manuscript addresses a significant clinical issue, atopic dermatitis (AD). The authors present findings on the application of Pulsatilla koreana Nakai (PKN) extract as a potential therapeutic agent for AD. The article is intriguing, novel, and aligns well with current research trends in exploring alternative treatments for inflammatory diseases. The study design is robust, and the results are presented in a clear and comprehensive manner.

Strengths:

  1. Novelty of the Topic: The exploration of alternative treatments for AD, particularly using medicinal plants, is highly relevant and holds potential clinical value.
  2. Methodology:
    • The study employs a comprehensive experimental model, incorporating both in vivo (NC/Nga mice) and in vitro (keratinocyte cell lines) analyses.
    • Molecular analyses, including western blotting, immunohistochemistry, and molecular studies, add credibility to the findings.
  3. Clarity of Results: The results are well-presented, with figures and tables supporting the central narrative effectively. Particularly valuable are the data on molecular mechanisms, such as the regulation of the JAK/STAT pathway by PKN.
  4. Practical Potential: The findings suggest that PKN extract could be utilized not only as a treatment for AD but also as an additive in functional foods to enhance skin health.

Comments and Suggestions for Improvement:

  1. Style and Language:
    • The manuscript contains several minor typographical errors and language issues, such as "dermstitia" instead of "dermatitis" and "siganling" instead of "signaling." For example in 130 line: “using hematoksylin and eosin”. A thorough linguistic and grammatical review is recommended.
    • Repetitions in the concluding sections should be minimized to improve readability and clarity.
  2. Methods Section:
    • The molecular docking procedure needs a more detailed description, including justification for the chosen experimental parameters.
    • Information on long-term monitoring of symptoms post-treatment in NC/Nga mice is lacking and should be included.
  3. Tables and Figures:
    • While the figures are well-prepared, more detailed captions would enhance their standalone value for readers.

4.     Discussion:

    • The results should be more extensively compared to findings from similar studies to emphasize the originality and significance of the work.
    • Limitations of the study, such as potential side effects of PKN in humans, should be discussed.
  1. References:
    • The citations are appropriate, but the inclusion of more recent on natural JAK/STAT inhibitors would strengthen the manuscript.

Conclusions:

The manuscript contributes significantly to research on alternative AD treatments, particularly through the use of Pulsatilla koreana Nakai extract. The findings are promising and warrant publication following the incorporation of the above suggestions. Enhancing the language quality and providing a more detailed discussion of the results and study limitations will further improve the manuscript’s impact.

Recommendation: Accept after minor revisions.

Author Response

  1. The manuscript contains several minor typographical errors and language issues, such as "dermstitia" instead of "dermatitis" and "siganling" instead of "signaling." For example in 130 line: “using hematoksylin and eosin”. A thorough linguistic and grammatical review is recommended. Repetitions in the concluding sections should be minimized to improve readability and clarity.

- Thank you for your comment. We revised the entire manuscript in relation to the parts you mentioned. We apologize for any inconvenience this may have caused you while reading, and we will make sure to check it more thoroughly before submitting.

  1. The molecular docking procedure needs a more detailed description, including justification for the chosen experimental parameters.

- We have added detailed description in the manuscript of the reasons for selecting the experimental parameters based on the reviewer's comments.

Revised manuscript: Exhaustiveness is a parameter that determines how thorough the docking analysis will be calculated. The default value is 8, and researchers can specify a number between 1 and 100 depending on their applications. The higher the number, the more rigorous the docking calculation will be. Previous studies have reported that there is little performance improvement when Exhaustiveness is greater than 25, but we set it to 100 for the most accurate analysis. Box_size was chosen as 126, the maximum value provided by the software, to consider all parts of the protein. (line 431)

  1. Information on long-term monitoring of symptoms post-treatment in NC/Nga mice is lacking and should be included

- Thank you for your comment. Neurological and physiological changes that may occur during the treatment process were monitored using indicators such as motor function, dietary intake, and body weight. This section was added to Material & Methods, 2.2.

  1. While the figures are well-prepared, more detailed captions would enhance their standalone value for readers.

- Thank you for your comment. We added captions based on what you said.

  1. The results should be more extensively compared to findings from similar studies to emphasize the originality and significance of the work.

- Thank you for your comment. We have taken your comments into consideration and added the following to the discussion part;

 "In particular, modern medical research on the efficacy and side effects of Baekduong is lacking. Most existing studies are based on traditional use cases or animal experiments, and there is a possibility of drug interactions when taken with other medications. Therefore, more safety evaluations and clinical studies are needed."

  1. The citations are appropriate, but the inclusion of more recent on natural JAK/STAT inhibitors would strengthen the manuscript.

- Thank you for your comment. We have taken your comments into consideration and added the following to the introduction part (line 70)

“Recent clinical trials and studies on atopic dermatitis have reported on topical JAK inhibitors such as ruxolitinib (JAK1/2) and delgocitinib (pan-JAK), as well as oral JAK inhibitors like baricitinib (JAK1/2) and upadacitinib (JAK-1 selective), suggesting the potential of JAK inhibitors as next-generation targeted therapies for AD.”

References

Chovatiya, R.; Paller, A. S., JAK inhibitors in the treatment of atopic dermatitis. J Allergy Clin Immunol 2021, 148, (4), 927-940.

Tampa, M.; Mitran, C. I.; Mitran, M. I.; Georgescu, S. R., A New Horizon for Atopic Dermatitis Treatments: JAK Inhibitors. J Pers Med 2023, 13, (3).

Reviewer 2 Report

Comments and Suggestions for Authors

The work presented to me for review certainly required a lot of work, a great many results were obtained, however, the authors did not respond to the results and the results of other authors in an appropriate manner in the discussion chapter. The chapter should be improved, the studied indicators and their role in AD should be discussed in the light of the available literature.

Consider dividing the in vitro and in vivo parts into two separate articles. In its current form, the work is not very legible/clear.

Author Response

  1. Consider dividing the in vitro and in vivo parts into two separate articles. In its current form, the work is not very legible/clear.

- Thank you for your thoughtful comment. First, I deeply regret that the work is readable. At the present, I’m having difficulty proceeding with the manuscript by dividing in to two parts.

So, I have revised the introduction, figure legends and discussion sections to make them more readable and clearer. I will take your suggestions into consideration and use them as a reference for further research to help make the writing process easier. Thanks again for your comments.

Reviewer 3 Report

Comments and Suggestions for Authors

The present paper evaluates effects of Pulsatilla Coreana treatment of dust mite-induced atopic dermatitis model in mice. The improvement of symptoms upon administration of P. Coreana extract in dose range 30-300 mg/kg has been observed via clinical examination (AD scoring and ear thickness), hystopathological examination of dorsal skin (dermal tickness and mast cell infiltration) and detection of inflammatory citokines and proteins involved in reintegration of skin barrier. The subsequent in vitro experiment with human keratynocites has confirmed the inhibitory effect of extract on JAK/STAT. The UPLC analysis of the extract has confirmed 7 potential active components. Docking into JAK/STAT proteins has revealed similar binding energies for all compounds into all proteins – range 7.8-11.5 kJ/mol.

The study is sound, the results are interesting and useful. I have several questions and suggestions

1. “The degree of dermatitis was evaluated every two weeks over a period of three weeks” – please explain. If possible, please provide a reference for the AD scoring.

2. Why have you selected dexasone as control treatment for AD and which dose has been applied?

3. Please provide structural formulae of UPLC-detected compounds.

4. Results of molecular docking are interesting, but similar for all involved components. Did you expect that? Have you considered “control docking” with some proven JAK/Stat inhibitor drug into the same proteins? Maybe you could drive out some more specific mechanistic insights.

Author Response

  1. “The degree of dermatitis was evaluated every two weeks over a period of three weeks” – please explain. If possible, please provide a reference for the AD scoring.

- Thank you for your comment. There seems to have been a slight issue with the description. I have revised the parts you mentioned and incorporated them into the manuscript. “~the degree of dermatitis was evaluated every twice a week for 3 weeks. (line 333)” Thank you for your checking. And references to AD scores have been added to the content.

References

Kim, H. J.; Song, H. K.; Park, S. H.; Jang, S.; Park, K. S.; Song, K. H.; Lee, S. K.; Kim, T., Terminalia chebula Retz. extract ameliorates the symptoms of atopic dermatitis by regulating anti-inflammatory factors in vivo and suppressing STAT1/3 and NF-kB signaling in vitro. Phytomedicine 2022, 104, 154318.

Shim, K. S.; Kim, H. J.; Ji, K. Y.; Jung, D. H.; Park, S. H.; Song, H. K.; Kim, T.; Kim, K. M., Rosmarinic Acid Ameliorates Dermatophagoides farinae Extract-Induced Atopic Dermatitis-like Skin Inflammation by Activating the Nrf2/HO-1 Signaling Pathway. Int J Mol Sci 2024, 25, (23).

  1. Why have you selected dexamethasone control treatment for AD and which dose has been applied?

- Thank you for your comment. Positive control drugs and doses were established based on the paper below and previous experimental results. The relevant contents were added to the M&M references. (line 313)

Reference

Mohd Kasim, V.N.K.; Noble, S.M.; Liew, K.Y.; Tan, J.W.; Israf, D.A.; Tham, C.L. Management of Atopic Dermatitis Via Oral and Topical Administration of Herbs in Murine Model: A Systematic Review. Front. Pharmacol. 2022, 13, 785782.

  1. Please provide structural formulae of UPLC-detected compounds.

- Thank you for your comment. I have revised the text by adding the section you pointed out to the image (line 198).

Round 2

Reviewer 2 Report

Comments and Suggestions for Authors

The corrections introduced had a minimal impact on the quality of the manuscript and did not contribute to its significant improvement. Despite the very extensive set of results, the discussion section remains underdeveloped. It lacks in-depth analysis and references to the literature. The discussion lacks reflection on the potential mechanisms behind the obtained results, as well as references to existing theories and hypotheses that could help in their interpretation. It is worth pointing out future directions of research that could fill the existing gaps in knowledge. Such additions would make the manuscript more comprehensive and valuable for readers.

Round 3

Reviewer 2 Report

Comments and Suggestions for Authors

Dear Authors,
Thanks for addressing the issues. The conclusions section should be under discussion.